# Genome-Wide Identification and Expression Analyses of the Cotton *AGO* Genes and Their Potential Roles in Fiber Development and Stress Response

**DOI:** 10.3390/genes13081492

**Published:** 2022-08-20

**Authors:** Mingchuan Fu, Yizhen Chen, Hao Li, Liguo Wang, Renzhong Liu, Zhanji Liu

**Affiliations:** Key Laboratory of Cotton Breeding and Cultivation in Huang-Huai-Hai Plain, Ministry of Agriculture and Rural Affairs, Institute of Industrial Crops Shandong Academy of Agricultural Sciences, Jinan 250100, China

**Keywords:** cotton, *AGO* gene family, expression pattern, Verticillium wilt, drought and salt stress

## Abstract

Argonaute proteins (AGOs) are indispensable components of RNA silencing. However, systematic characterization of the *AGO* genes have not been completed in cotton until now. In this study, cotton *AGO* genes were identified and analyzed with respect to their evolution and expression profile during biotic and abiotic stresses. We identified 14 *GaAGO*, 14 *GrAGO*, and 28 *GhAGO* genes in the genomes of *Gossypium arboreum*, *Gossypium raimondii*, and *Gossypium hirsutum*. Cotton AGO proteins were classified into four subgroups. Structural and functional conservation were observed in the same subgroups based on the analysis of the gene structure and conserved domains. Twenty-four duplicated gene pairs were identified in *GhAGO* genes, and all of them exhibited strong purifying selection during evolution. Moreover, RNA-seq analysis showed that most of the *GhAGO* genes exhibit high expression levels in the fiber initiation and elongation processes. Furthermore, the expression profiles of *GhAGO* genes tested by quantitative real-time polymerase chain reaction (qPCR) demonstrated that they were sensitive to Verticillium wilt infection and salt and drought stresses. Overall, our results will pave the way for further functional investigation of the cotton *AGO* gene family, which may be involved in fiber development and stress response.

## 1. Introduction

Argonaute (AGO) proteins are core effectors of the RNA-induced silencing complex (RISC), which are well known to regulate gene expression in RNA interference (RNAi) or RNAi-independent pathways [1]. For instance, Arabidopsis AGO10 specifically binds to microRNA166/165 to regulate the development of the shoot apical meristem. A loss of function of *AGO10* resulted in the pinhead phenotype in the Arabidopsis ecotype Ler mutant plants [2]. Typically, AGO proteins contain four domains viz. N-terminal (Argo-N), Piwi Argonaute Zwille (PAZ), MID, and PIWI domains. The Argo-N domain is believed to participate in the separation of the small RNA:target duplex, while the PAZ domain contains a specific binding pocket that can anchor small RNAs. The MID domain can bind the 5′ phosphates of small RNAs and anchors small RNAs onto the AGO proteins. The PIWI domain is functionally similar to RNase H with endonuclease activity and is in charge of the cleavage of target mRNA. In addition to these domains, the Argo-L1 and Argo-L2 domains are revealed in a number of AGO proteins [3].

The *AGO* gene family has been widely studied in a number of plant species. In general, different numbers of *AGO* genes are present in various plant species. Ten *AGO* genes have been revealed in *Arabidopsis thaliana* [4], 19 in *Oryza sativa* [5], 17 in *Zea mays* [6], 15 in *Solanum lycopersicum* [7], 13 in *Vitis vinifera* [8], 7 in *Cucumis sativus* [9], 27 in *Brassica napus* [10], 12 in *Capsicum annuum* [11], 13 in *Citrus sinensis* [12], 69 in *Triticum aestivum* [13], 13 in *Phaseolus vulgaris* [14], 14 in *Solanum tuberosum* [15], 18 in *Camellia sinensis* [3], and 13 in *Musa acuminata* [16]. However, no *AGO* genes in *Gossypium* species have been elucidated to date.

*Gossypium hirsutum* (AD_1_) is an allotetraploid, which evolved from interspecific hybridization between an A-subgenome species and a D-subgenome species around 1–2 million years ago [17]. *G. arboreum* (A_2_) and a *G. raimondii* (D_5_) are considered as the putative donor diploid species for the A- and D-subgenomes, respectively. Cotton has been grown as the most important natural fiber crop and provides approximately 35% of the total fiber consumed worldwide [18]. It is also a vital oilseed and feed crop in the world owing to the high contents of oil and protein in cottonseeds [19]. However, cotton production is severely constrained by several biotic and abiotic stresses such as Verticillium wilt, salt, and drought, which result in decreased yields and inferior harvest quality. In this study, members of the *AGO* gene family were identified from systematic analyses of the allotetraploid *G. hirsutum* genome, as well as two diploid progenitors *G. arboreum* and *G. raimondii*. The gene structure and conserved domains of these genes were comprehensively analyzed, and the gene duplication events were identified. The expression patterns of *GhAGO* genes were examined in various organs using public RNA-Seq data and after treatments with Verticillium wilt, salt, and drought via quantitative RT-PCR (qPCR). This study will provide comprehensive information about the *AGO* genes and pave the way for further investigation of their function in cotton.

## 2. Materials and Methods

### 2.1. Identification of Gossypium AGO Genes

*A. thaliana AGO* genes were retrieved from the Arabidopsis Information Resource (TAIR v10, https://www.arabidopsis.org/) (accessed on 19 November 2021) [20], while rice (*O. sativa* L., v7.0) *AGO* genes were obtained from the Phytozome v13 (https://phytozome-next.jgi.doe.gov/) (accessed on 19 November 2021) [21]. We used the HMMsearch program (http://hmmer.org/) (accessed on 22 November 2021) to screen the genome sequences of *G. hirsutum* (HAU, v1.1), *G. arboreum* (CRI, v1.0), and *G. raimondii* (JGI, v2.0) (https://cottonfgd.net/) (accessed on 22 November 2021) [22], employing the hidden Markov model (HMM) profiles of the PAZ domain (PF02170) and PIWI domain (PF02171) downloaded from the Pfam database (http://pfam.xfam.org/) (accessed on 22 November 2021) as queries [23]. The resulting sequences were then verified by the Pfam and Conserved Domain Database (CDD, https://www.ncbi.nlm.nih.gov/cdd) (accessed on 22 November 2019) [24]. The ExPASy ProtParam tool (https://web.expasy.org/protparam/) (accessed on 23 November 2021) was used to predict the molecular weight (MW) and theoretical isoelectric point (pI) of each AGO protein. Subcellular localization analysis was performed using the web server BUSCA (http://busca.biocomp.unibo.it/) (accessed on 23 November 2021) [25].

### 2.2. Phylogenetic, Gene Structure, and Conserved Domain Analyses

The phylogenetic tree was built by the neighbor-joining (NJ) method using MEGA 11 [26]. Bootstrap analysis was simulated with 1000 replicates. The exon–intron structures were visualized using the Gene Structure Display Server 2.0 (GSDS 2.0, http://gsds.cbi.pku.edu.cn/) (accessed on 22 March 2022) [27]. Conserved domain analysis of the AGO proteins was performed by the HMMER web server (https://www.ebi.ac.uk/Tools/hmmer/search/hmmscan) (accessed on 22 March 2022) using default parameters [28].

### 2.3. Chromosomal Mapping and Gene Duplication Analyses

The positions of the *GhAGO* genes were determined by mRNA location information retrieved from the GFF files of *G. hirsutum* genome. Additionally, the chromosomal distribution of *GhAGO* genes was illustrated by the MapChart (v2.32) program [29]. Gene duplication events were determined using methods described previously [30]. The Circos program was used to demonstrate the relationships of duplicated genes [31]. The value of nonsynonymous substitutions to synonymous substitutions (Ka/Ks) was calculated using the KaKs_Calculator package [32].

### 2.4. Cis-Acting Regulatory Element Analysis

We extracted the upstream 1500 bp DNA sequences of the *GhAGO* genes from the *G. hirsutum* genome sequences [33] and, then, submitted to the PlantCARE database (http://bioinformatics.psb.ugent.be/webtools/plantcare/html/) (accessed on 8 April 2022) [34]. The bed-files obtained from the above database were illustrated using GSDS 2.0 [27].

### 2.5. Transcriptome Data Analysis

The tissue-specific expression patterns of upland cotton *AGO* genes were measured using the transcriptome data of *G. hirsutum* TM-1, which were retrieved from NCBI (https://www.ncbi.nlm.nih.gov/bioproject/PRJNA248163) (accessed on 11 April 2022) [35]. The heatmap of the 28 *GhAGO* genes’ expression profiles were performed using the TBtools package [36].

### 2.6. Plant Materials and Stress Treatments

Healthy seeds of *G. hirsutum* cv. Lumian418 were planted in sterilized soil at 28 °C with a photoperiod of 16 h light/8 h dark. Seedlings were grown for two weeks, then gently transferred to Hoagland solution for two days, and finally, transferred to Hoagland solution containing 200 mM NaCl or 15% (*w*/*v*) PEG6000. For Verticillium wilt infection, two-week-old seedlings were infected by the high-virulence VD8 strain of *Verticillium dahliae* (2 × 10^7^ spores/mL) by the root-dip method [37]. Roots from three biological replicates were harvested at 2, 6, and 12 h after treatment.

### 2.7. Quantitative Real-time PCR Analysis

RNA was isolated using Trizol reagent (Invitrogen, Carlsbad, CA, USA) and digested with RNase-free DNase I (Takara, Dalian, China) to eliminate trace genomic DNA. The quality of RNA samples was investigated by 1% agarose gel electrophoresis, while the concentration was examined by a spectrophotometer (NanoDrop 2000, Waltham, MA, USA). First-strand cDNA was synthesized from 1 µg of RNA using a PrimerScript^TM^ 1st Strand cDNA Synthesis Kit (Takara, Dalian, China). The qPCR assay was performed in an ABI QuantStudio 5 Real-Time PCR System (Thermo Fisher Scientific, Waltham, MA, USA) using SYBR^®^ Premix Ex Taq^TM^ (Takara, Dalian, China) with three replicates. The qPCR procedure was set as follows: one cycle at 95 °C for 3 min, then 40 cycles at 95 °C for 15 s, and 60 °C for 15 s. The cotton *Histone3* (AF024716) was used as the internal reference gene [38]. The relative expression levels were calculated according to the 2^−ΔΔCt^ method [39]. The qPCR primers are listed in Appendix A.

### 2.8. Statistical Analysis

The qPCR data from three biological replicates were analyzed as the mean ± the standard error. The difference between treatment and control was evaluated by Tukey’s honestly significant difference tests. “*” and “**” indicate significant differences at *p* ≤ 0.05 and *p* ≤ 0.01, respectively.

## 3. Results

### 3.1. Identification and Phylogenetic Analysis of Gossypium AGO Genes

To determine the *AGO* genes in the genomes of *G. raimondii*, *G. arboretum*, and *G. hirsutum*, we carried out a genome-wide survey using HMMER search with the PAZ domain (PF02170) and PIWI domain (PF02171). After manual inspection and confirmation using the Pfam and CDD databases, 14 *G. raimondii AGO* (*GrAGO*), 14 *G. arboretum AGO* (*GaAGO*), and 28 *G. hirsutum AGO* (*GhAGO*) genes were identified in total (Appendix A). The nomenclature of cotton *AGO* genes was defined based on the closest orthologs in Arabidopsis and rice. Comparison analysis suggested that the *G. hirsutum* genome harbored all copies of the *AGO* genes from the two diploid progenitor species. In addition, we found that the length of *Gossypium* AGO proteins ranged from 359 (GhAGO5c) to 1145 (GaAGO1b) amino acids. The predicted molecular weight ranged from 40.93 to 127.31 kDa, and the calculated pI values ranged from 8.98 to 9.61 (Appendix A). Additionally, all *Gossypium* AGO proteins were predicted to be nuclear proteins, which were highly consistent with AGO proteins from Arabidopsis and rice.

To identify the evolutionary relationship of the *Gossypium* AGO proteins, we constructed a phylogenetic tree using the NJ method (Figure 1). The result indicated that the *Gossypium* AGO proteins clustered into four separate subgroups (i.e., AGO1, MEL1, AGO4, and ZIPPY), which was consistent with Arabidopsis, rice, and pepper [11]. Among the four subgroups, subgroup AGO1 contained the most AGO members with 23 *Gossypium* AGO proteins, while subgroup MEL1 contained the least AGO members with only 9 *Gossypium* AGO proteins. In addition, both AGO4 and ZIPPY subgroups contained 12 *Gossypium* AGO proteins. Furthermore, compared with those in Arabidopsis, *AGO1*, *AGO4*, *AGO5*, *AGO7*, and *AGO10* have greatly expanded in *Gossypium* species, while *AGO3*, *AGO8*, and *AGO9* were not detected in cotton.

### 3.2. Genomic Localization and Gene Duplication Analysis of GhAGO Genes

*GhAGO* genes were anchored to their corresponding chromosomes (Figure 2). A total of 27 *GhAGO* genes unevenly distributed on 16 chromosomes and one (*GhAGO5e*) was mapped on a D-subgenome scaffold region (Scaffold635). Among the 16 chromosomes, 11 chromosomes (A06, A07, A08, A10, A12, D06, D07, D08, D10, D12, and D13) contained a single *GhAGO* gene. In addition, 2 chromosomes (A05 and A13) and 3 chromosomes (A09, D05, and D09) possessed two and four *GhAGO* genes, respectively (Figure 2).

To reveal *GhAGO* duplication events, we performed a genome-wide collinearity analysis using the MCScanX program [40]. As a result, 2159 collinear blocks were identified in the *G. hirsutum* (HAU) genome and 54,839 genes (78.12%) were collinear genes. In particular, 22 pairs of *GhAGO* genes were segmental duplications, which involved 22 *GhAGO* genes, whereas two pairs (*GhAGO5a*/*GhAGO5b* and *GhAGO5c*/*GhAGO5d*) were tandem duplications (Figure 3 and Appendix A). The Ka/Ks values of all duplication pairs were less than 1, ranging from 0.063 to 0.859 (Appendix A), indicating that the *AGO* gene family in *G. hirsutum* had undergone purifying selection during the long evolutionary period.

### 3.3. Gene Structure and Conserved Domain Analysis of GhAGO Genes

We analyzed the *GhAGO* gene structure and display it in Figure 4. The number of exons in the *GhAGO* genes varied from 3 in *GhAGO2d*, *GhAGO7a*, *GhAGO7b*, and *GhAGO7c* to 26 in *GhAGO5a*, which might be related to the diversification of their functions. Interestingly, the *GhAGO* genes from the same phylogenetic subgroup share highly similar gene structures. For example, in the subgroup AGO4, most *GhAGO* genes and the counterparts from *G. arboretum* and *G. raimondii* contained 21 introns and 22 exons. However, *GhAGO6d* had 22 introns and 23 exons. Additionally, *GhAGO5c* showed a distinct pattern of gene structure. It seemed to have lost part of the nucleotides from both the PAZ and Piwi domains.

Seven conserved domains were identified among the 28 GhAGO proteins (Figure 4). As a result, with the exception of GhAGO5c, all GhAGO proteins shared four conserved domains, N-terminal ArgoN domain, PAZ, ArgoL1, and C-terminal Piwi domain, which is in line with known plant AGOs [41,42]. Notably, a Gly-rich Ago1 domain was revealed in front of the ArgoN domain in the GhAGO1a/b/c/d/e proteins. Additionally, previous efforts revealed that the Piwi domain exhibited substantial homology to RNase H and enabled some AGO proteins to cleave target RNAs pairing to the bound small RNAs [43]. Generally, the catalytic activity is associated with the conserved triad (aspartate-aspartate-histidine, DDH) and an additional conserved histidine at position 798 (H798) in Arabidopsis AGO1 [5,8]. In this study, we aligned the Piwi domains of all GhAGO proteins and 10 AtAGO proteins using ClustalX (http://www.clustal.org/, accessed on 22 November 2021). The result indicated that 17 GhAGO proteins contained the conserved DDH/H798 residues (Appendix A). Among the other 11 GhAGO proteins, 6 GhAGOs, all from the AGO4 subgroup, contained the conserved DDH triad, but the histidine at the 798th position in AtAGO1 was replaced by a proline. In GhAGO2a and GhAGO2d, the histidine at the 986th position in AtAGO1 was replaced by an aspartate.

### 3.4. Cis-Acting Regulatory Elements in Promoter Region of GhAGO Genes

A number of *cis*-acting regulatory elements were revealed in the promoter regions of the 28 *GhAGO* genes. The essential regulatory elements such as TATA-box and CAAT-box were detected in all *GhAGO* genes. Other *cis*-acting elements identified in the *GhAGO* genes can be divided into four groups according to their functional properties, namely light response, stress response, plant growth, and hormone-responsive elements. The distribution of these elements identified in promoter regions of each *GhAGO* gene is illustrated in Figure 5. In particular, we identified eight elements associated with six hormone responses. These *cis*-acting elements include the AuxRR-core and TGA-element related to auxin response, the GARE-motif and P-box associated with gibberellin response, ABRE involved in abscisic acid (ABA) response, ERE related to ethylene-response, the CGTCA-motif involved in methyl jasmonate (MeJA) response, and the TCA-element associated with salicylic acid (SA) response. Additionally, the promoter sequences of some *GhAGO* genes also contain several elements involved in environmental stress responses, including pathogen defense (AT-rich and TC-rich), cold (DRE and LTR), wounding (WUN-motif), and anaerobic stress (ARE). Taken together, these results suggested that the *GhAGO* genes might play vital roles in regulating cotton response to hormone and environmental stresses.

### 3.5. GhAGO Gene Expression Patterns in Diverse Cotton Tissues

The expression patterns of *GhAGO* genes in diverse tissues were investigated by using the method described by Zheng et al. [44]. The results indicated that the expression levels of the *GhAGO* genes were significantly different in diverse tissues (Figure 6). *GhPNH1a* and *GhPNH1b* were detected with high expression levels in the leaf and ovule 20 days post-anthesis (dpa). *GhAGO5c* showed a relatively distinctive expression pattern compared to all the other analyzed genes. This was due to the significant difference in its level of accumulation in the pistil. Additionally, all of the *GhAGO* genes showed extremely low expression levels in the calycle, petal, and stamen (Figure 6). In particular, most of the *GhAGO* genes were expressed highly in the early (−3–5 dpa) and middle (3–20 dpa) stages of ovule development, suggesting that these genes may function in the fiber initiation and elongation processes.

### 3.6. GhAGO Genes Were Influenced by Verticillium Wilt Infection

To investigate whether the *GhAGO* genes play roles in pathogen defense in cotton, we analyzed the transcriptional patterns of the *GhAGO* genes in response to *V. dahliae* infection by qPCR at 2, 6, and 12 h post-inoculation (hpi). The results demonstrated that Verticillium wilt infection significantly affected the expression of the *GhAGO* genes (Figure 7). Half of the tested *GhAGO* genes were significantly suppressed by Verticillium wilt infection, while five *GhAGO* genes (*GhAGO2a*, *GhAGO4a*, *GhAGO7c*, *GhAGO7d*, and *GhPNH1d*) were remarkably induced by 2h treatment with Verticillium wilt infection. In addition, five members out of the tested *GhAGO* genes contained pathogen-responsive element AT-rich or TC-rich. Verticillium wilt infection significantly influenced the expression levels of *GhAGO2d*, *GhAGO7a*, *GhPNH1a*, and *GhPNH1d*, but had almost no effect on the expression of *GhAGO1a* (Figure 7). Interestingly, some *GhAGO* genes might have undergone neofunctionalization after duplication. For instance, the expression level of *GhAGO7a* was significantly downregulated by 2h treatment with *V. dahliae* infection, while its homeolog, *GhAGO7d*, was greatly upregulated (Figure 7).

### 3.7. GhAGO Genes Were Modulated by Salt and Drought Stresses

Accumulating evidence indicates that RNA silencing exerts essential functions in plant resistance to abiotic stresses [11,13]. Therefore, the expression of the *GhAGO* genes was evaluated after salt and drought treatments. The qPCR assay showed that salt and drought stresses resulted in significant differences in the expression levels of the *GhAGO* genes (Figure 8). Surprisingly, all of the tested *GhAGO* genes were significantly downregulated by salt stress, indicating that these *GhAGO* genes might be important in response to salt stress. Similarly, the expression levels of seven *GhAGO* genes were downregulated by drought stress. In addition, the expression of *GhAGO1a* decreased significantly at 2 h after drought treatment, increased remarkably at 6 h, and then, decreased at 12 h. The expression level of *GhAGO10b* was greatly upregulated at 2 h and downregulated at 6 h. Conversely, the expression levels of *GhAGO2a*, *GhAGO2d*, and *GhAGO7c* were not modified under drought stress (Figure 8).

## 4. Discussion

### 4.1. Characterization of Gossypium AGO Genes

In this study, we carried out a survey of the *Gossypium AGO* genes at the whole-genome scale to examine their potential functions in fiber development and stress response. Consequently, 28 *GhAGO* genes were identified in the *G. hirsutum* genome, which is twice as much as that found in *G. arboreum* and *G. raimondii*, presumably because *G. hirsutum* is a tetraploid species evolved from the hybridization between the diploid *G. raimondii* and *G. arboretum* [17]. Specifically, the respective number of *AGO2*, *AGO4*, *AGO6*, *AGO7*, *AGO10*, and *PNH1* genes in the tetraploid *G. hirsutum* is exactly double the numbers in diploid cotton, suggesting that no recent gene duplication or deletion occurred in these *AGO* genes after allotetraploid formation. However, gene deletion in *AGO1* and gene duplication in *AGO5* were observed because tetraploid upland cotton contains 5 *AGO1* and 5 *AGO5* genes, while each diploid cotton contains 3 *AGO1* and 2 *AGO5* genes, respectively (Figure 1).

Cotton *AGO* genes were classified into four subgroups viz. AGO1, MEL1, AGO4, and ZIPPY, which is consistent with the previous results based on 206 *AGO* genes from 23 plant species [45]. The *AGO* genes, however, were divided into three subgroups (AGO1/5/10, AGO4/6/8/9, and AGO2/3/7) in several plant species such as common bean [14] and potato [15]. Indeed, compared to three subgroups in common bean and potato, we split the subgroup AGO1/5/10 into two subgroups AGO1 and MEL1 (homolog of AGO5) with well-supported bootstrap values, which is consistent with the results reported for rice [5] and grapevine [8]. Extensive studies showed that several AGO subgroups have expanded through lineage-specific duplication [45]. For instance, compared with Arabidopsis, subgroups AGO1 and MEL1 (AGO5) expanded in a number of flowering species. More specifically, common bean harbors four *AGO10* genes [14], whereas wheat contains 15 *AGO5* paralogs [13]. In cotton, expansion of *AGO5* and *AGO10* has been also identified. The upland cotton contains five *AGO5* and six *AGO10* paralogs (Figure 1). In addition, *AGO1* (five members), *AGO4* (four members), and *AGO7* (four members) have significantly expanded in the *G. hirsutum* genome. Meanwhile, we further investigated *AGO* gene duplication in the upland cotton genome and revealed 24 duplicated *GhAGO* gene pairs, including 22 segmental duplication pairs and 2 tandem duplication pairs (Figure 3). These results suggested that segmental duplication was the primary driving force for the expansion of the *GhAGO* genes. Similarly, the expansion of wheat *AGO* genes was dominated by segmental duplication [13]. Furthermore, all duplicated gene pairs had undergone strong purifying selection in the process of evolution (Appendix A), indicating that purifying selection exerted an important role in the formation of the *GhAGO* gene functions. Previous studies have demonstrated that the cotton lineage experienced a five- to sixfold ploidy increase approximately 57–70 million years ago (Mya) [46], while the A and D ancestor genome diverged around 6.2−7.1 Mya [17]. In this study, about half of the duplication events occurred after the cotton lineage ploidy increase. Additionally, one duplicate (*GhAGO7a*/*GhAGO7d*) might occur after the divergence of the two diploid progenitors (Appendix A).

### 4.2. Differential Expression of GhAGO Genes in Response to Multiple Stresses

Extensive evidence shows that *AGO2* and *AGO4* play vital roles in the modulation of plant immunity [47,48,49]. In Arabidopsis, *AtAGO2*, which was highly induced by the bacterial pathogen *Pseudomonas syringae*, regulates innate immunity by binding miRNA393b^*^ to orchestrate exocytosis of antimicrobial pathogenesis-related proteins [47]. *AtAGO4* acts as a positive regulator of DNA methylation and mediates resistance to *P. syringae* [48]. In *Nicotiana attenuata*, among all 11 *NaAGO* genes, only transcripts of *NaAGO4a* and *NaAGO4b* were induced by hemibiotrophic pathogen *Fusarium brachygibbosum* infection. A loss of function of *NaAGO4* confers mutant hypersusceptible to *F. brachygibbosum* [49]. Moreover, a strong upregulation of *PvA_AGO2a* and *PvA_AGO4a* expression was observed in *P. vulgaris* after inoculation with the fungus *Colletotrichum lindemuthianum* [14]. Additionally, *StAGO15* in *Solanum tuberosum* belonging to the AGO4 clade was suppressed at 0–3 dpi and then significantly activated at 4–5 dpi when *Phytophthora infestans* had completed the transition from the biotrophic to necrotrophic stage [15]. In this study, to determine the contribution of some of the *GhAGO* genes involved in the defense response in upland cotton, we performed expression analysis in roots inoculated with the fungus *V. dahliae* based on the qPCR assay. The expression levels of *GhAGO2a*, *GhAGO4a*, *GhAGO7c*, *GhAGO7d*, and *GhPNH1d* were significantly upregulated at 2 hpi. On the contrary, the expression of *GhAGO1b*, *GhAGO2d*, *GhAGO5b*, *GhAGO5d*, *GhAGO7a*, *GhAGO7b*, and *GhAGO10b* was downregulated (Figure 7). Notably, the transcripts of most of the *GhAGO* genes containing pathogen-responsive element were significantly influenced by *V. dahliae* infection. These results suggest that these *GhAGO* genes may participate in the regulation of cotton defense.

Emerging evidence suggests that the *AGO* genes not only contribute to biotic stress response, but also modulate plant resistance to abiotic stresses such as salt, drought, cold, and heat stresses [8,13,50]. Arabidopsis *AtAGO2* was significantly induced by salt stress. Further analysis revealed that AtAGO2 improves Arabidopsis salt tolerance by interacting with an R3H-type RNA binding protein MUG13.4 and then influences the SOS signaling cascade at the transcription level [51]. In *Z. mays*, 17 *ZmAGO* genes have been reported, and all *ZmAGO* genes were induced under drought stress. Surprisingly, transcripts of *ZmAGO18a* and *ZmAGO18b* were 539.9-fold and 730.8-fold upregulated at 1h under drought stress in comparison to the control. The mutation of *ZmAGO18b* rendered plants hypersensitive to drought stress [50]. In *Setaria italica*, the mutation of *SiAGO1b* resulted in enhanced susceptibility to drought stress [52]. In addition, the expression levels of most of the *TaAGO* genes in *T. aestivum* were influenced by salt and drought stresses [13]. Similarly, most of the *VvAGO* genes in *V. vinifera* were downregulated under salt and drought stresses [8]. Salt and drought are considered the most destructive abiotic stresses to cotton. In this study, all of the tested *GhAGO* genes were significantly suppressed by salt stress. Two *GhAGO* genes (*GhAGO1a* and *GhAGO10b*) exhibited upregulation in response to drought stress at specific time points. This result indicated that these *GhAGO* genes may play important roles in plant adaptation to salt and drought stresses.

## 5. Conclusions

This study performed a systematic analysis of the *AGO* gene family in three *Gossypium* species. A total of 14 *GrAGO*, 14 *GaAGO*, and 28 *GhAGO* genes were revealed in the genomes of *G. raimondii*, *G. arboretum*, and *G. hirsutum*, respectively. The *Gossypium AGO* genes were divided into four distinct subgroups. Duplication analysis demonstrated that the *GhAGO* genes experienced segmental and tandem duplication events during evolution. Furthermore, the predicted *cis*-acting regulatory elements of the *GhAGO* genes suggested their functional association with growth, development, hormone response, and environmental stress response. Tissue-specific expression analysis indicated that most of the identified *GhAGO* genes may play pivotal roles in the fiber initiation and elongation processes. Our qPCR analyses revealed that a number of *GhAGO* genes were involved in the response to *V. dahliae* infection and salt and drought stresses. Overall, our results will provide a solid foundation for further functional characterization of the *GhAGO* genes in response to biotic and abiotic stresses.

## Figures and Tables

**Figure 1 genes-13-01492-f001:**
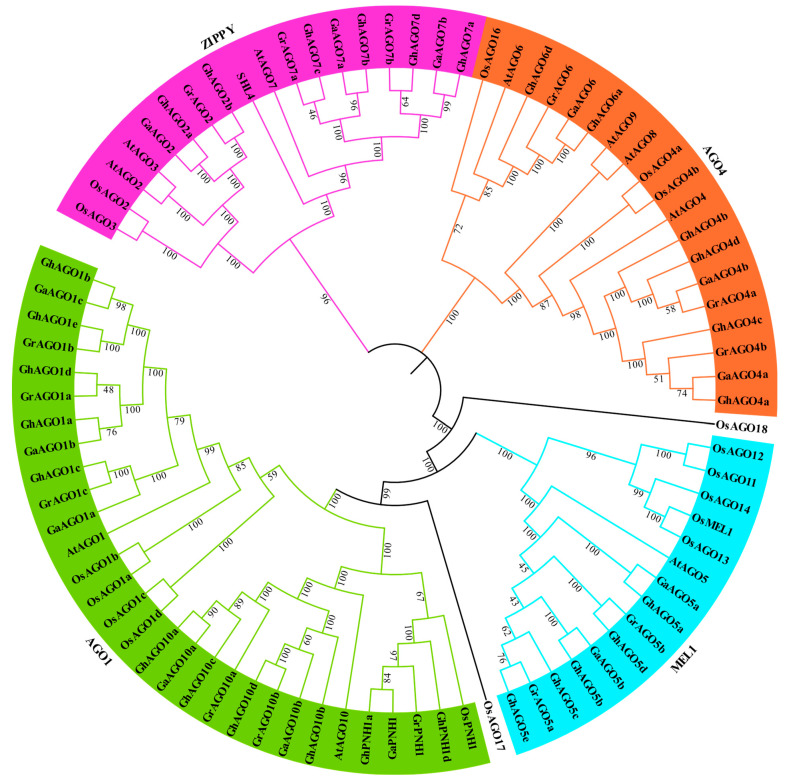
Phylogenetic tree of the Gossypium AGO proteins. The AGO protein sequences of Gossypium, Arabidopsis, and rice were aligned using ClustalW. The MEGA 11 program was used to generate the NJ tree with 1000 bootstrap replicates. Different subgroups of AGO proteins are highlighted with various colors.

**Figure 2 genes-13-01492-f002:**
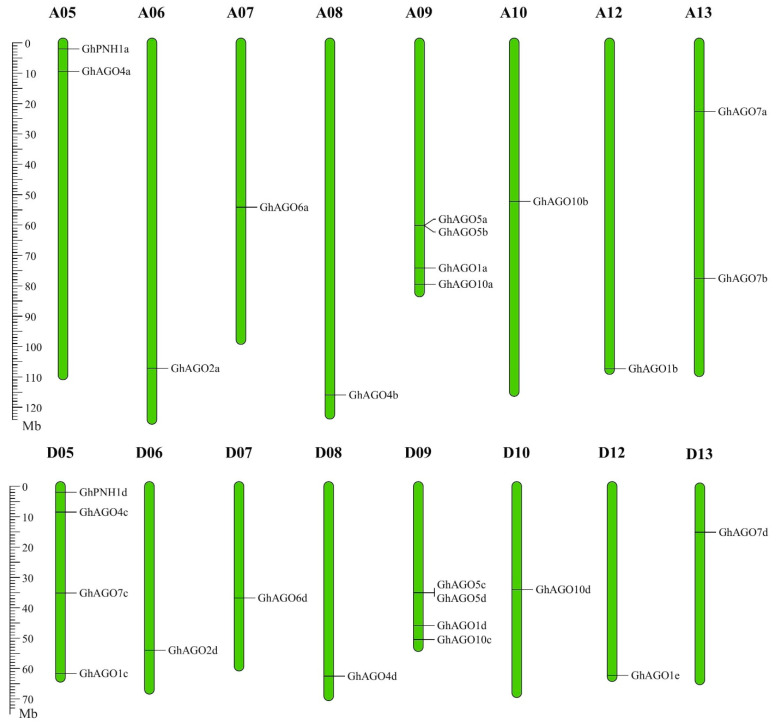
Chromosomal localizations of the *GhAGO* genes. Green bars indicate the *G. hirsutum* chromosomes. The scale bar on the left denotes the chromosomal lengths (Mb).

**Figure 3 genes-13-01492-f003:**
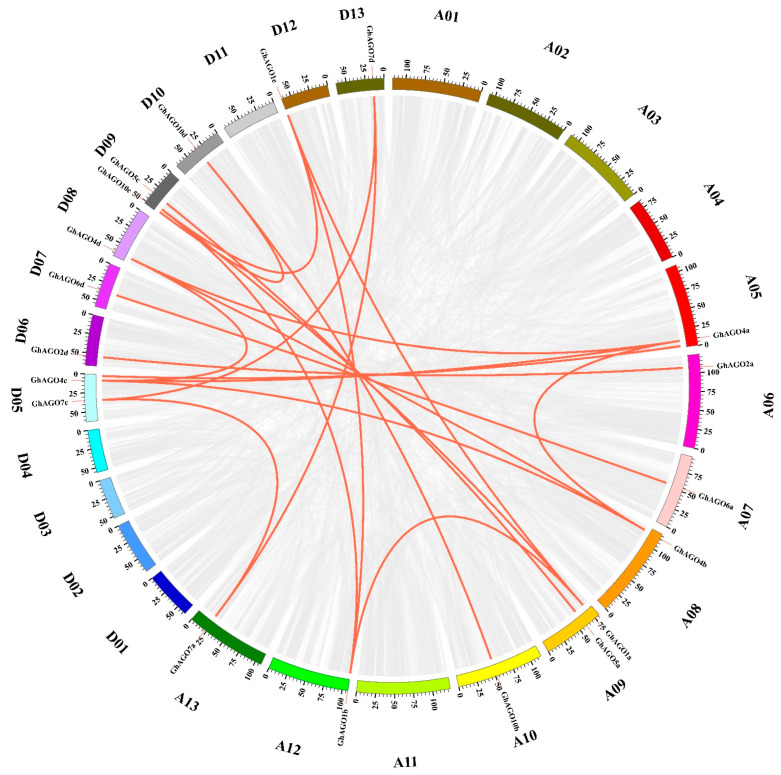
Collinearity analysis of *GhAGO* genes in *G. hirsutum*. The red lines highlight the 22 pairs of segmental duplications. The scale bar marked on the chromosome indicates chromosome lengths (Mb).

**Figure 4 genes-13-01492-f004:**
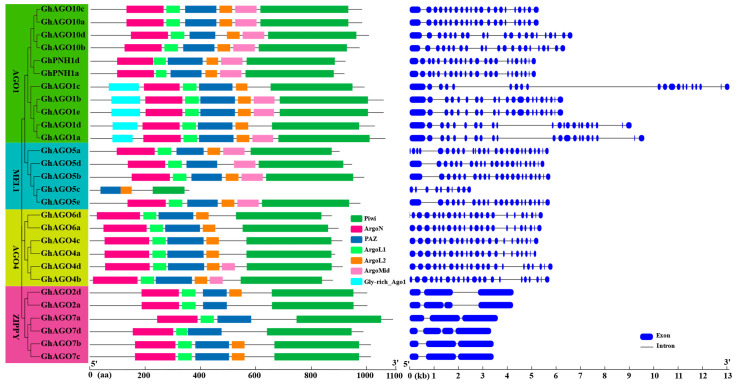
Conserved domains and gene structures of the *GbNAC* genes. A: Phylogenetic tree of the *GhAGO* genes. Multiple sequence alignment of the GhAGO proteins was performed using ClustalW. The neighbor-joining (NJ) tree was constructed using MEGA 11 with 1000 bootstrap replicates. B: Conserved domain of the *GhAGO* genes. The conserved domain of the GhAGO proteins was detected by the HMMER web server (https://www.ebi.ac.uk/Tools/hmmer/search/hmmscan) (accessed on 22 March 2022) using default parameters. C: Gene structure of the *GhAGO* genes. The blue boxes and black lines represent exons and introns, respectively.

**Figure 5 genes-13-01492-f005:**
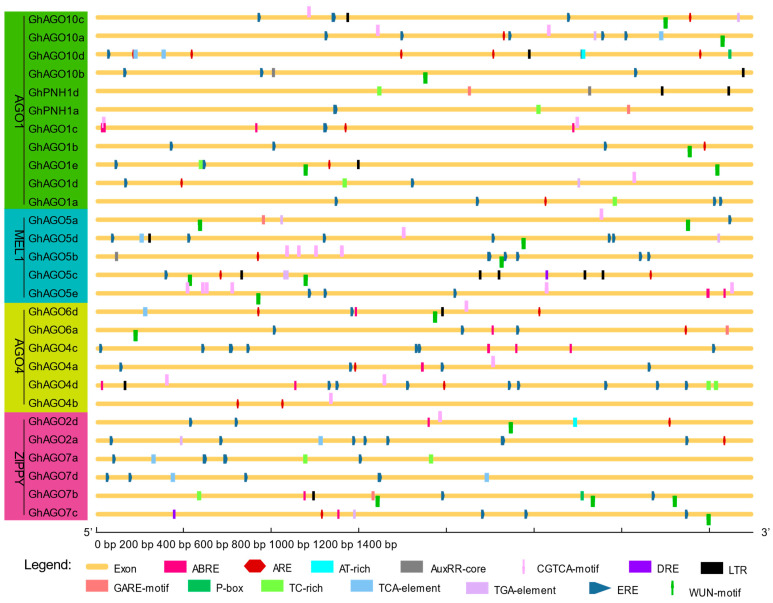
Analysis of cis-acting regulatory elements of the *GhAGO* genes. The yellow lines represent the promoter regions of the *GhAGO* genes. The scale bar at the bottom denotes the length of the promoter sequence.

**Figure 6 genes-13-01492-f006:**
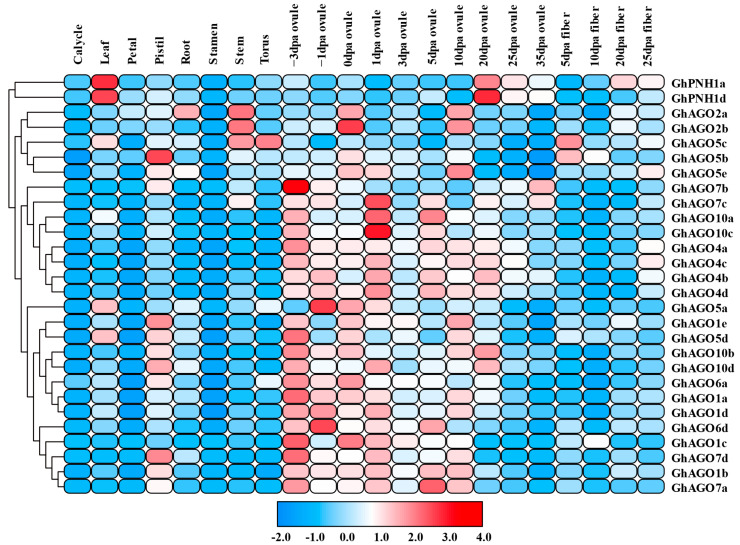
Expression profiles of the *GhAGO* genes in various tissues. The expression levels are illustrated in different colors on the scale. Red represents high expression, and blue indicates low expression. dpa is an abbreviation for days post-anthesis.

**Figure 7 genes-13-01492-f007:**
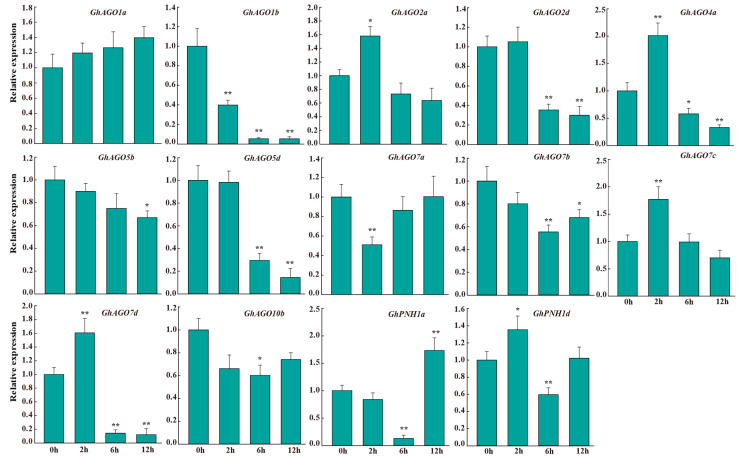
Expression analyses of the *GhAGO* genes in roots under Verticillium wilt infection. qPCR was used to determine the expression profiles of 14 *GhAGO* genes under Verticillium wilt infection. The standard deviation is indicated by the error bars, and “*” (Tukey’s HSD, *p* ≤ 0.05) and “**” (*p* ≤ 0.01) indicate significant differences between the treatment and control.

**Figure 8 genes-13-01492-f008:**
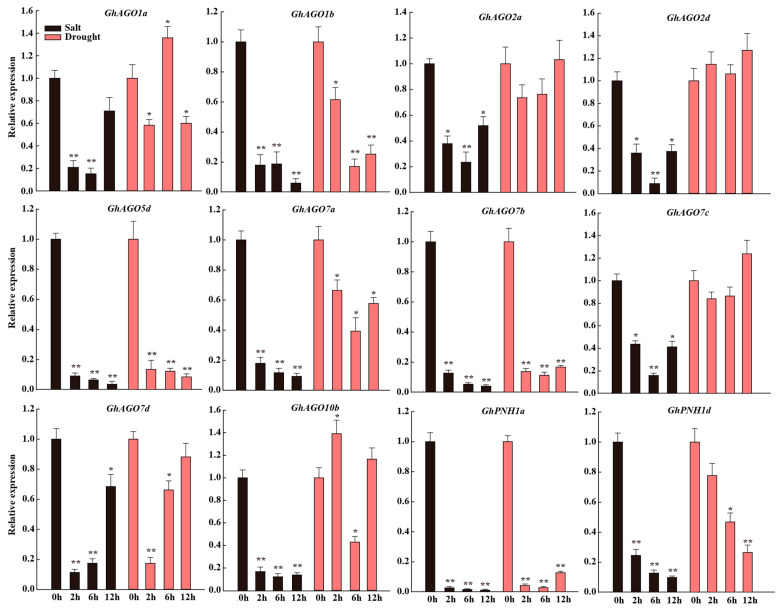
Expression analyses of the *GhAGO* genes in roots under salt and drought stresses. qPCR was used to determine the expression profiles of 12 *GhAGO* genes under salt and drought stresses. The standard deviation is indicated by the error bars, and “*” (Tukey’s HSD, *p* ≤ 0.05) and “**” (*p* ≤ 0.01) indicate significant differences between the treatment and control.

## Data Availability

Not applicable.

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
