# Peer review of "Genome-Wide Identification and Expression Analyses of the Cotton AGO Genes and Their Potential Roles in Fiber Development and Stress Response"

_genes, 2022, doi:10.3390/genes13081492_

Round 1

Reviewer 1 Report

Authors perfomed interesting study related to phylogenetic, bioinformatic and gene expression (transcriptomic, RT-PCR) analysis of Argonaute gene family in the three cotton species. Research is generally well planned and realized. However, there is a problem with the access to supplementary data and data presentation. Some major comments should be addressed to further improve the manuscript.

1. Line 10- should be Argonaute

2. Paragraph 2.7

Provide the length of PCR products, method of RNA quality assessment.

3. Figure 5. The signs describing cis-active elements are not unambiguous. Particularly those for GARE-GATA, TGACG-ERE-DRE, TGA-CGTCA-CAG. Authors use rectangles in different colors, that could be not a bad solution at a small number of signs. However, here is over 20 of them. Maybe the solution could be the use not only rectangles but also balls, triangles or squares. Even when colors could be related, the shape will help in the proper cis-elements discrimination.

4. Authors should describe statistical methods, for example in a separate paragraph in materials and methods.

5. Authors show RT-PCR results, suggesting that GhAGO genes were influenced by Verticillium wilt as well as were sensitive to salt and drought treatment.

Authors should also state in results section which cis-active elements in particular gene promoters could be responsible for presented RT-PCR results.  Add also several related sentences into the discussion.

6. There is problem with the access to Supplementary material- assure that all supplementary data (Figures, Tables etc.)  are correctly added.

Author Response

Authors performed interesting study related to phylogenetic, bioinformatic and gene expression (transcriptomic, RT-PCR) analysis of Argonaute gene family in the three cotton species. Research is generally well planned and realized. However, there is a problem with the access to supplementary data and data presentation. Some major comments should be addressed to further improve the manuscript. 

Response: Thanks for the nice comments on our manuscript.

1. Line 10- should be Argonaute

Response: Thanks. We changed the “Argonaut” in Line 10 to “Argonaute”.

2. Paragraph 2.7

Provide the length of PCR products, method of RNA quality assessment.

Response: Thanks for the good suggestion. We added the data in supplementary file Table S1 and relevant descriptions on page 3, lines 116-118.

3. Figure 5. The signs describing cis-active elements are not unambiguous. Particularly those for GARE-GATA, TGACG-ERE-DRE, TGA-CGTCA-CAG. Authors use rectangles in different colors, that could be not a bad solution at a small number of signs. However, here is over 20 of them. Maybe the solution could be the use not only rectangles but also balls, triangles or squares. Even when colors could be related, the shape will help in the proper cis-elements discrimination.

Response: Thanks for the valuable comments. In order to show clearly the signs describing cis-acting elements, we illustrated the 14 regulatory elements involved in hormone responses and environmental stress responses. The solution was improved by using rectangle, wedge, double sided wedge, and arrow.

4. Authors should describe statistical methods, for example in a separate paragraph in materials and methods.

Response: Thanks for your suggestions. We added the statistical methods and relevant descriptions on page 3 lines 125-129.

5. Authors show RT-PCR results, suggesting that GhAGOgenes were influenced by Verticillium wilt as well as were sensitive to salt and drought treatment.

Authors should also state in results section which cis-active elements in particular gene promoters could be responsible for presented RT-PCR results.  Add also several related sentences into the discussion.

Response: Thanks for your valuable comments. We added relevant description on page 10, lines 261-264, and page 12, lines 351-353.

6. There is problem with the access to Supplementary material- assure that all supplementary data (Figures, Tables etc.)  are correctly added.

Response: Thanks. We uploaded all the supplementary files including five tables.

Reviewer 2 Report

I checked your manuscript and described comments below.

I think this paper is a good analysis of cotton argonaut genes and proteins.

But I think the problem is in the following part.

1.       Pytosome is now v13. Why did you use v11?

2.       The following database URLs are different.

Phytozome: https://phytozome-next.jgi.doe.gov/

CottenFDG: https://cottonfgd.net/

Gossypium hirsutum cultivar:Texas Marker-1 (TM-1) (cotton): https://www.ncbi.nlm.nih.gov/bioproject/PRJNA248163

3.       Why did you use MEGA X for phylogenetic tree analysis, but now MEGA11 is the new software.

4.       Figure 1 does not have bootstrap values. You should write bootstrap values.

5.       Missing supplementary files. You should upload those files.

I don't think this paper has any major mistakes or grammatical problems.

Author Response

I checked your manuscript and described comments below.

I think this paper is a good analysis of cotton argonaut genes and proteins.

Response: Thanks for the good comments on our manuscript.

But I think the problem is in the following part.

1. Pytosome is now v13. Why did you use v11?

Response: Thanks. The rice genome (v7.0) was retrieved from the Phytozome database (v11) a couple of years ago. The rice genome has not been updated since 2007. We changed the Phytozome v11 to Phytozome v13 (Page 3, line 68).

2. The following database URLs are different.

Phytozome: https://phytozome-next.jgi.doe.gov/

CottenFDG: https://cottonfgd.net/

Gossypium hirsutum cultivar:Texas Marker-1 (TM-1) (cotton): https://www.ncbi.nlm.nih.gov/bioproject/PRJNA248163

Response: Thanks for your valuable suggestions. We rechecked all the URLs mentioned in our manuscript and changed the URLs of the Phytozome (Page 3, line 68), CottonFGD (Page 3, line 71), and the transcriptome data of G. hirstutum TM-1 (Page 4, line 102). Actually, the CottonFGD has two URLs viz. https://www.cottonfgd.org/ and https://cottonfgd.net/. The https://www.cottonfgd.org/ is under maintenance right now.

3. Why did you use MEGA X for phylogenetic tree analysis, but now MEGA11 is the new software.

Response: Thanks. We downloaded the latest version of MEGA (v11) and reconstructed the phylogenetic tree of cotton AGO genes. No difference was observed between the phylogenetic tree constructed by MEGA X and MEGA 11.

4. Figure 1 does not have bootstrap values. You should write bootstrap values.

Response: Thanks for your comments. We added the bootstrap values to Figure 1 by using the MEGA 11 software.

5. Missing supplementary files. You should upload those files.

Response: Thanks. We uploaded all the supplementary files including five tables.

I don't think this paper has any major mistakes or grammatical problems.

Response: Thanks for the good comments.

Round 2

Reviewer 1 Report

Authors properly addressed all issues presented in the review round 1. The article can be publised in present form .

Author Response

Authors properly addressed all issues presented in the review round 1. The article can be published in present form.

Response: Thanks for your approval of our revised manuscript.

Reviewer 2 Report

I checked your revised manuscript and described previous comments below.

1.       Pytosome is now v13. Why did you use v11?

I confirmed the fix.

2.       The following database URLs are different.

Phytozome: https://phytozome-next.jgi.doe.gov/

CottenFDG: https://cottonfgd.net/

Gossypium hirsutum cultivar:Texas Marker-1 (TM-1) (cotton): https://www.ncbi.nlm.nih.gov/bioproject/PRJNA248163

I confirmed the fix in all of the above.

3.       Why did you use MEGA X for phylogenetic tree analysis, but now MEGA11 is the new software.

I confirmed the fix.

4.       Figure 1 does not have bootstrap values. You should write bootstrap values.

I confirmed the fix.

5.       Missing supplementary files. You should upload those files.

I checked the supplementary file. I think that there is no particular problem with the content.

I have checked all the fixes.

Author Response

I checked your revised manuscript and described previous comments below.

1. Pytosome is now v13. Why did you use v11?

I confirmed the fix.

2. The following database URLs are different.

Phytozome: https://phytozome-next.jgi.doe.gov/

CottenFDG: https://cottonfgd.net/

Gossypium hirsutum cultivar:Texas Marker-1 (TM-1) (cotton): https://www.ncbi.nlm.nih.gov/bioproject/PRJNA248163

I confirmed the fix in all of the above.

3. Why did you use MEGA X for phylogenetic tree analysis, but now MEGA11 is the new software.

I confirmed the fix.

4. Figure 1 does not have bootstrap values. You should write bootstrap values.

I confirmed the fix.

5. Missing supplementary files. You should upload those files.

I checked the supplementary file. I think that there is no particular problem with the content.

I have checked all the fixes.

Response: Thanks for your approval of the revised manuscript.